# A Greedy Hierarchical Approach to Whole-Network Filter-Pruning in CNNs

**Kiran Purohit**                                   *kiran.purohit@kgpian.iitkgp.ac.in*
**Anurag Reddy Parvathgari**                        *anuragparvathgari7@gmail.com*
**Sourangshu Bhattacharya**                         *sourangshu@cse.iitkgp.ac.in*
*Department of Computer Science and Engineering*
*Indian Institute of Technology, Kharagpur, India*

**Reviewed on OpenReview:** *https: // openreview. net/ forum? id= WzHuebRSgQ*

## Abstract

Deep convolutional neural networks (CNNs) have achieved impressive performance in many computer vision tasks. However, their large model sizes require heavy computational resources, making pruning redundant filters from existing pre-trained CNNs an essential task in developing efficient models for resource-constrained devices. Whole-network filter pruning algorithms prune varying fractions of filters from each layer, hence providing greater flexibility. State-of-the-art whole-network pruning methods are either computationally expensive due to the need to calculate the loss for each pruned filter using a training dataset, or use various heuristic / learned criteria for determining the pruning fractions for each layer. Hence there is a need for a simple and efficient technique for whole network pruning. This paper proposes a two-level hierarchical approach for whole-network filter pruning which is efficient and uses the classification loss as the final criterion. The lower-level algorithm (called filter-pruning) uses a sparse-approximation formulation based on linear approximation of filter weights. We explore two algorithms: orthogonal matching pursuit-based greedy selection and a greedy backward pruning approach. The backward pruning algorithm uses a novel closed-form error criterion for efficiently selecting the optimal filter at each stage, thus making the whole algorithm much faster. The higher-level algorithm (called layer-selection) greedily selects the best-pruned layer (pruning using the filter-selection algorithm) using a global pruning criterion. We propose algorithms for two different global-pruning criteria: (1) layerwise-relative error (HBGS), and (2) final classification error (HBGTS). Our suite of algorithms outperforms state-of-the-art pruning methods on ResNet18, ResNet32, ResNet56, VGG16, and ResNext101. Our method reduces the RAM requirement for ResNext101 from 7.6 GB to 1.5 GB and achieves a 94% reduction in FLOPS without losing accuracy on CIFAR-10.

## 1 Introduction

Convolutional neural networks (CNNs) have demonstrated remarkable performance across various applications, such as image classification (Han et al., 2016), object detection (Redmon et al., 2016), and image segmentation (Minaee et al., 2021). However, the deployment of CNNs on IoT devices for computer vision tasks often encounters practical bottlenecks related to the model size and computational complexity of inference (FLOPs) (Goel et al., 2020). While neural architecture search (Baker et al., 2017; Zoph & Le, 2017) and efficient model design (Tan & Le, 2019) can sometimes lead to highly efficient architectures, they impose substantial requirements in terms of data and computational cost, as well as research expertise. However, pruning of pre-trained models (Lebedev & Lempitsky, 2018; Hoefler et al., 2021; Vadera & Ameen, 2022; He & Xiao, 2023) provides a cheaper alternative where one can avoid re-training complicated models on large datasets. For CNNs, *structured pruning* or *filter-pruning* (FP) (He et al., 2017; Luo et al., 2017; He & Xiao, 2023) has emerged as a preferred alternative since it causes a reduction in computation (thus leading to power savings) as well as memory requirements without requiring special hardware or re-implementation of operations.

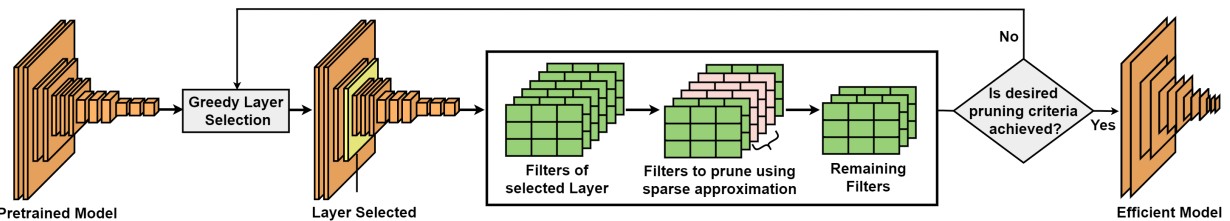

Figure 1: Hierarchical approach for non-uniform pruning of filters across the network.

Filter-pruning (FP) techniques can further be classified as (1) *layer-wise pruning*, which prune filters uniformly from each layer (e.g. score-propagation (Yu et al., 2018) and error in activation reconstruction (Luo et al., 2017)), and (2) *whole-network pruning* (WNP), which prunes filters from the entire network. The WNP approach can prune different fractions of filters from each layer, hence providing higher flexibility. An important challenge for WNP is to determine the pruning fractions for each layer. (Kuzmin et al., 2019) (section 3.4.1) calculates the accuracy by pruning individual layers to a varying fraction and finds an optimal compromise so that the overall pruning ratio is achieved while minimizing the maximum loss in accuracy per layer. The main disadvantage of this approach is that the effect of pruning one layer on the pruning of another layer is not captured. Recent works also include methods based on Taylor-series expansion of the loss function, which approximates the influence score of pruning a filter on the overall loss function (Wang et al., 2019; Molchanov et al., 2019; Peng et al., 2019; Nonnenmacher et al., 2021) with good practical performance (He & Xiao, 2023). However, these methods can be expensive for large networks as they require passing over the entire training set to calculate each influence score, which can be costly for large datasets. Additionally, (Dong & Yang, 2019) applied NAS to search for a network with flexible channel and layer sizes, but this method can also be expensive for larger networks. On the other hand, some recent works use approximate criteria to prune filters. For instance, (Murti et al., 2023) propose a discriminative pruning technique based on total variation separation distance (TVS), which is an approximate criterion to prune filters from a network. Similarly, (He et al., 2020) choose different criteria to prune filters from different layers using Gumbel-softmax. However, the main drawback of this procedure is that the Gumbel-softmax smoothing only calculates an approximate output feature map for each layer, thus potentially hurting overall performance. Therefore, there is a need for an *efficient* and *accurate* WNP technique that directly optimizes the *training data loss*.

In this work, we propose a greedy hierarchical training data loss-based approach for whole-network filter pruning (see fig. 1). The iterative higher-level algorithm (called *layer-selection*) evaluates all layers based on outputs from the lower-level algorithm, and greedily selects a layer to prune filters from in each iteration. The lower-level algorithm (called *filter-pruning*) prunes filters optimally for the current network configuration. We propose two versions of the iterative layer-selection algorithm: (1) **hierarchical backward greedy search (`HBGS`)**, which selects layers based on the relative reconstruction error of the layer outputs, and (2) **hierarchical backward greedy tree search (`HBGTS`)** which selects the layers based on the error of the final classification layer outputs. The key advantage of our greedy layer-selection, compared to a learned criterion (He et al., 2020), or a threshold-based criterion (Kuzmin et al., 2019) is that we utilize the activations from the modified network, which arguably leads to better decisions after a few iterations of pruning. However, since each iteration of the greedy layer-selection makes many calls to the filter-pruning algorithm (typically, the number of layers many calls, with some possibility of caching old results), an expensive filter-pruning algorithm would be impractical for large networks.

A key contribution of this paper is to propose an *efficient* filter-pruning algorithms, which can ensure the feasibility of the overall hierarchical scheme for large networks. While LRF (Joo et al., 2021) demonstrated impressive practical performance for pruning filters, it only prunes one filter at-a-time, hence making it prohibitively expensive for our purpose. We formulate the problem of optimally pruning multiple filters from a layer using linear replaceability criteria as a sparse approximation problem. We study an orthogonal matching pursuit (OMP) (Tropp & Gilbert, 2007) based algorithm, FP-OMP (Purohit et al., 2023) for filter-pruning. Under the assumption of restricted isometry of the matrix composed of filter weights (Tropp & Gilbert, 2007), FP-OMP selects filters whose linear combinations can represent the pruned filters with

minimal error. However, since FP-OMP follows a greedy forward selection algorithm, and is called iteratively to prune a small number of filters, the overall algorithm becomes computationally expensive. To alleviate this inefficiency, we propose `FP-Backward` – a backward elimination-based algorithm for solving the sparse approximation problem. A key facilitating factor towards a fast implementation of `FP-Backward` is the calculation of a closed-form expression for the approximation error incurred by pruning a filter. `HBGTS` along with `FP-Backward` (called `HBGTS-B`) is an *efficient* algorithm taking only 50% of the running time of `HBGTS` and with performance comparable to it.

Experimental results on a variety of standard pre-trained CNN models, e.g. ResNet18, ResNet32, ResNet56, VGG16, ResNext101 on standard image classification datasets, e.g. CIFAR10, CIFAR100, Tiny-Imagenet show that models pruned with `HBGS` and `HBGTS` have higher accuracies compared to recent state-of-the-art pruning methods for similar compression levels (see Table 1 and Figure 2). At higher parameter reduction ($\geq$90%), the proposed methods outperform existing methods by $\sim 5\%$ (see Figure 2). We also find the optimal pruned model to have a highly non-uniform pruning fraction distribution for each layer (see Figure 4), hence showing the effectiveness of our layer-selection algorithm. To summarize:

1. We propose a novel *greedy hierarchical framework* for non-uniform pruning of filters with *filter-pruning* at the lower level and *layer-selection* at a higher level.

2. We propose a backward-elimination-based scheme, `FP-Backward` for filter pruning, which takes advantage of a novel closed-form expression for approximation error.

3. We propose `HBGTS` which uses an efficient implementation to directly optimize the classification error for layer selection.

## 2   Related Work

Many pruning methods have been proposed in the literature. (Hoefler et al., 2021; Vadera & Ameen, 2022) provide excellent surveys for pruning techniques. Pruning can be categorised two types: *unstructured pruning*, involving the removal of individual weights (Han et al., 2015), and *structured pruning* or *filter-pruning* (FP), in which entire nodes or channels are removed (He et al., 2017; Luo et al., 2017; He & Xiao, 2023). Structured pruning provides efficiently implementable models on a wide range of accelerator devices e.g. GPUs. (He & Xiao, 2023) provides a recent survey and website for comparing structured pruning techniques for CNNs. Pruning can be done on a pre-trained model or from scratch, which is costly and requires large training data. Therefore we focus on pruning a pre-trained model. We further categorise it into the following groups:

**Weight-Based Pruning -** Weights of filters are examined to determine which ones are essential for the model's performance. These methods do not require input data. (Han et al., 2015) focused on eliminating small-norm weights. (He et al., 2019) incorporates geometric median (Fletcher et al., 2008) to estimate the importance of each filter. (Joo et al., 2021) prunes linearly replaceable filters.

**Activation-Based Pruning -** Rather than relying on filter weights, these methods utilize activation maps or layer outputs to make pruning decisions. We can utilize information from activation maps at the *current layer* or *all layers/whole-network*. Some of the *current layer* activation-based pruning methods are: CP (He et al., 2017) which focuses on minimizing the reconstruction error of sparse activation maps, while HRank (Lin et al., 2020) calculates the average rank of activation maps. CHIP (Sui et al., 2021) assesses cross-channel correlation to evaluate channel importance. ThiNet (Luo et al., 2017; El Halabi et al., 2022) approximates activation maps of layer l+1 using subsets of layer l's activation maps. *Current layer* activation-based pruning methods do not consider the reconstruction error propagation. Some of the *all layers/whole-network* activation-based pruning methods are: NISP (Yu et al., 2018) which assesses the Final Response Layer to calculate the neuron importance, while DCP (Zhuang et al., 2018) aims to retain discriminative channels. Layer-output based methods are computationally expensive, since they need to compute the outputs using a training dataset.

**Regularization -** Regularization can aid in learning structured sparse networks by incorporating various sparsity regularizers. These regularizers can be implemented on *Batch Normalization (BN) parameters* (Liu et al., 2017; You et al., 2019; Kang & Han, 2020), with *extra parameters* (Huang & Wang, 2018; Lin et al., 2019) and *filters* (Wen et al., 2016; Chen et al., 2021).

**Taylor Expansion -** Taylor Expansion is employed to approximate the change in the loss by pruning filters. *First-order-Taylor* uses the first-order information to calculate the loss change caused by pruning weights (Molchanov et al., 2019; You et al., 2019). *Second-order-Taylor* exploits the Hessian matrix, containing second-order information (Peng et al., 2019; Wang et al., 2019; Liu et al., 2021). However, these methods can be expensive for large networks.

**Coreset-Based Pruning -** Recently, several methods (Tukan et al., 2022; Liebenwein et al., 2019; Mussay et al., 2021) have utilized the concept of coresets for pruning DNNs. These approaches focus on providing bounds on the approximation error incurred at each layer of prediction. However, they do not necessarily achieve optimal pruning in terms of representation error. Tukan et al. (2022) employs an upper bound on sensitivity for sampling filters, which can result in overly pessimistic sampling weights. On the other hand, our method focuses on removing linearly redundant filters, which is done optimally for a given layer.

There is a different line of work in which pruned models are effectively trained from *scratch* e.g. Frankle & Carbin (2018); Rachwan et al. (2022). Unlike training-based approaches, our method does not require training from scratch, which is costly and requires large training data.

## 3 A Hierarchical Greedy Approach to Filter Pruning

We propose a Hierarchical scheme, `HBGS/HBGTS` for non-uniform filter pruning from a pre-trained CNN. As shown in Figure 1, the proposed scheme operates in a two-level hierarchical manner: (1) *filter pruning* - at the lower level, this step identifies the most appropriate filters to be pruned from each layer and (2) *layer selection* - at a higher level this step selects the best layer to currently prune from. These two steps are applied iteratively to achieve a non-uniform pruning from the whole network. We first describe our sparse approximation-based formulation for optimal filter pruning from each layer, and then describe a faster backward elimination-based algorithm for the same. For layer selection, we describe a layerwise-regression-based backward greedy search strategy. We also incorporate an overall error-based strategy for layer selection.

### 3.1 Sparse Approximation for Filter Pruning

A convolutional filter used in deep CNN is denoted by a $K \times K$ matrix. A convolutional layer is defined as filter weights $f_{i,j} \in \mathbb{R}^{K^2}$, where $i = 1, ..., m$ and $j = 1, ..., n$ are the number of input and output channels. Given the input feature map with $m$ channels $X = \{X_1, ..., X_m\}$, the output feature map with $n$-channels $Y = \{Y_1, ..., Y_n\}$, can be calculated as: $Y_j = \sum_{i=1}^{m} X_i * f_{i,j} := X * f_{:,j}$

Here, $*$ denotes the convolution operation, and $f_{:,j} \in \mathbb{R}^{K^2 \times m}$ denotes all the filter weights for output channel $j$. For brevity, we describe the algorithm for output channel pruning throughout the paper. Input channel pruning can be performed analogously. For channel pruning, we follow the idea of *linearly replaceable filters* (LRF) introduced in (Joo et al., 2021), which states that any filter $f_{:,j} \in \mathbb{R}^{K^2 m}$ can be pruned if the filter weights can be expressed as a linear combination of other filter weights of the same layer which are not pruned. Note that, for linear approximation of a filter with respect to other filters of the same layer, we treat the filter weights, $f_{:,j}$, as a flat $K^2 m$-dimensional vector. For LRF, we prune the channel $j$ such that $||\epsilon_j||$ is minimum, where $f_{:,j} = \sum_{l \neq j} \lambda_{j,l} f_{:,l} + \epsilon_j$. Here, $\lambda_{j,l}$ are the coefficients of $l^{th}$ filter for approximating the $j^{th}$ filter, which can be computed by solving the minimization problem: $\min_{\lambda_{j,:}} ||f_{:,j} - \sum_{l \neq j} \lambda_{j,l} f_{:,l}||^2$. LRF (Joo et al., 2021) works by iteratively pruning one filter using the above technique, and updating the weights of the retained parameters using one epoch of SGD updates minimizing a combination of training loss and knowledge distillation loss (Hinton et al., 2015) w.r.t. to unpruned model outputs.

The above method can be generalized by selecting a set of filters, $S$, in one go. Given the set of selected filters $S$, the error for $j^{th}$ output filter, $f_{:,j} \notin S$ is given by $\epsilon_j$:

$$f_{:,j} = \sum_{l \in S} \lambda_{j,l} f_{:,l} + \epsilon_j, \forall j \notin S \tag{1}$$

The problem of estimating $\lambda_{j,l}, l \in S$ can be posed as a sparse approximation problem:

$$S^*, \lambda^* = \operatorname{argmin}_{|S| \leq (1-\beta)n, \lambda} \sum_{j \in \{1,2,..,n\}} ||f_{:,j} - \sum_{l \in S} \lambda_{j,l} f_{:,l}||^2 \tag{2}$$

---

**Algorithm 1** Filter Pruning-OMP (FP-OMP)

1: **Input:** $n$: Number of filters,
2: $\quad\quad\quad \beta$: Pruning fraction,
3: $\quad\quad\quad f_{:,j} \in \mathbb{R}^{K^2 m} \;\; j = 1, ....n$: Filters
4: **Initialize:**
5: $\quad$ Normalize $f_{:,j}$ such that $||f_{:,j}||_2 = 1$
6: $\quad R_j = f_{:,j} \;\; \forall j \in \{1, 2, .., n\}$ ▷ Residual error
7: $\quad S = \phi$ $\quad\quad\quad\quad$ ▷ Set of selected filters
8: **while** $|S| \leq (1 - \beta) * n$ **do**
9: $\quad$ **for** $i$ in $S^c$ **do**
10: $\quad\quad$ **for** $j$ in $\{1, 2, .., n\}$ **do**
11: $\quad\quad\quad$ Compute $Proj_{ij} = R_j.f_{:,i}$
12: $\quad\quad$ **end for**
13: $\quad\quad$ Total projection $\xi_i = \sum_{j=1}^{n} |Proj_{ij}|$
14: $\quad$ **end for**
15: $\quad ind = \max_i \xi_i$
16: $\quad S \longleftarrow S \cup \{ind\}$
17: $\quad$ **for** $j$ in $\{1, 2, .., n\}$ **do**
18: $\quad\quad \vec{\lambda}_{j,:} = argmin_{\lambda_j} ||f_{:,j} - \sum_{l \in S} \lambda_{j,l} f_{:,l}||^2$
19: $\quad\quad R_j \longleftarrow f_{:,j} - \sum_{l \in S} (\vec{\lambda}_{j,:}.f_{:,l})$
20: $\quad$ **end for**
21: **end while**
22: **Output:**
23: $\quad S, \lambda_{j,l} \;\; \forall l \in S \;\; \forall j \in \{1, 2, .., n\}$

---

**Algorithm 2** Layer Selection: `HBGS`

1: **Input:** $C$: Number of layers,
2: $\quad\quad F_c \;\; c = 1, ..., C$: Filters,
3: $\quad\quad \mathcal{D}$: Training dataset,
4: $\quad\quad \alpha$: Number of filters pruned in one go,
5: $\quad\quad \beta$: Total pruning ratio over all layers
6: **Initialize:** $y_c^0 = F_c^0 * y_{c-1}^0 \;\; c = 1, ..., C, \; t \leftarrow 0$
7: **while** Overall pruning ratio $< \beta$ **do**
8: $\quad e_c^t = 0 \;\; \forall c = 1, ..., C$
9: $\quad G_c^t \leftarrow$ FP-OMP$(n, \frac{\alpha}{n}, F_c^t) \;\; \forall c = 1, ..., C$
$\quad$ where $n = |F_c^t|$
10: $\quad$ **for** $i = 1, ..., |D|$ **do**
11: $\quad\quad$ **for** $c = 1, ..., C$ **do**
12: $\quad\quad\quad$ Calculate output $y_c^t(i) = F_c^t * y_{c-1}^t(i)$
13: $\quad\quad\quad e_c^t = e_c^t + \frac{||y_c^0(i) - G_c^t * y_{c-1}^t(i)||_2}{||y_c^0(i)||_2}$
14: $\quad\quad$ **end for**
15: $\quad$ **end for**
16: $\quad cmin = \arg\min_c e_c$
17: $\quad$ Revised network params $F_{cmin}^{t+1} = G_{cmin}^t$ ;
$\quad F_c^{t+1} = F_c^t \;\forall c \neq cmin$
18: $\quad t \leftarrow t + 1$
19: $\quad$ Run 1 epoch of finetuning.
20: **end while**
21: **Output:** Pruned filters $F_c^t \;\; \forall c = 1, ..., C$

---

where $n$ is the initial number of output channels in the current layer, and pruning ratio $\beta$, is the fraction of channels that are pruned. Algorithm 1 describes an *orthogonal matching pursuit* (OMP) based approximation (Tropp & Gilbert, 2007; Cai & Wang, 2011) for estimating the $S, \lambda_{j,l} \;\; \forall j = 1, ..., n; \; l \in S$. Note that, equation 2 denotes a multi-variate regression version of the sparse approximation problem where the predicted variable is a vector $f_{:,j}, j = 1, ..., n$ with corresponding independent parameters $\lambda_{j,:}$. Since the total error is the sum of squared errors of the individual components, it is easy to see that projections used in standard OMP algorithm can be replaced with the sum of projections for each multivariate regression component (line 13 of Algorithm 1). This approach has two advantages: (1) this approach is much faster than LRF since the fine-tuning is performed once for each layer, whereas in LRF it is performed after every pruning step (which is equal to the number of pruned filters in a layer), and (2) this approach provides an optimality guarantee for the selected filters in terms of reconstruction error, under conditions on the incoherence matrix of the features (Cai & Wang, 2011). The overall time complexity of algorithm 1 is $\mathcal{O}(|S|n^3)$. In a normal application scenario of uniform pruning, the pruning fraction $\beta$ may be quite high ($\sim 98\%$), resulting in the size of the selected set $|S|$ being much smaller than $n$, this algorithm is fast ($\mathcal{O}(n^3)$).

LRF also uses a $1 \times 1$ convolution layer $g_{j,k}, j, k = 1, ..., n$ to compensate for the loss of channel outputs. The modified output feature map, $Z_k, k = 1, ..., n$ is given by $Z_k = \sum_{j=1}^{n} Y_j * g_{j,k} := \sum_{j=1}^{n} X * f_{:,j} * g_{j,k}$, when the output filters $Y_j, j = 1, ..., n$ are not pruned. However, after pruning, the output feature map from the original convolutional layer becomes $Y_j' = \sum_{l \in S} X * f_{:,l}$. *Weight compensation* is a method for modifying weights for the $1 \times 1$ convolutional layer, $g_{l,k}', l \in S, k = 1, ..., n$ such that the final predicted output $Z_k' = \sum_{l \in S} X * f_{:,l} * g_{l,k}'$ matches $Z_k$. The following result provides a formula for calculating $g_{l,k}'$.

**Result 1.** *Given $Z_k$, $Z_k'$, $g_{j,k}$, and $g_{l,k}'$ defined as above, and $\lambda_{j,l}, j = 1, ..., n; \; l \in S$ estimated using the filter pruning process. Letting $g_{l,k}' = g_{l,k} + \sum_{l' \in S^c} \lambda_{l',l} * g_{l',k}, \; \forall l \in S, \; k = 1, ..., n$, ensures that $Z_k - Z_k' = \sum_{l' \in S^c} X * \epsilon_{l'} * g_{l',k}$, where $\epsilon_{l'}$ is the error vector for the estimation of removed filter $l' \in S^c$, and $S^c$ denotes the set of all removed filters.*

For brevity, the derivation of the result is provided in the appendix. This result provides us with a formula for updating the weights of the $1 \times 1$ filters, thus obviating the need to update them using the SGD update.

## 3.2 Hierarchical Backward Greedy Search (`HBGS`)

The algorithm outlined in the previous section selects a fixed fraction $(1 - \beta)$ of filters from each layer. However, as shown in the results, each layer can have a different fraction of important filters, depending on the architecture. Hence, determining the fraction of filters $\beta_c$ to be pruned in layer $c$ is an important problem. Intuitively, $\beta_c$ should not be determined by the filter-weights since comparing them across layers is not meaningful. For example, the weights in a layer may be scaled by a constant factor, compared to those in another layer. Hence, we use reconstruction error of filter outputs using input training dataset as the criteria.

Let $\mathcal{D} = \{(u_1, v_1), ..., (u_N, v_N)\}$ be the training dataset, and $y_c(i)$ be the output feature map of layer $c$ when the training datapoint $(u_i, v_i)$ is input to the CNN. Also, let $U_c(i)$ be the output of the $c^{th}$ layer when the training datapoint $(u_i, v_i)$ is input to the unpruned CNN. Moreover, let $F_c = \{\sum_{l' \in S_c} (f_{:,l'}^c * g_{l',k}^c), \forall k = 1, ..., n_c\}$ be the composite convolutional map of the pruned filters and $1 \times 1$ convolution for layer $c$ obtained from a filter pruning method (e.g. FP-OMP described in the previous section). The relative reconstruction error $e_c$ for layer $c$ is given by: $e_c = \sum_{(u_i, v_i) \in \mathcal{D}} \frac{||U_c(i) - F_c * y_{c-1}(i)||_2}{||U_c(i)||_2}$. We propose a *hierarchical backward greedy search* (`HBGS`) technique in algorithm 2 to both estimate $\beta_c$ for each layer $c$, as well as select the appropriate filters from each layer. Given the number of filters $\alpha$ to be pruned in one go, the algorithm proceeds iteratively by performing two broad steps in each iteration: (1) determine the optimal $\alpha$ filters to be pruned in each layer $c$, and (2) calculate the total relative reconstruction error $e_c$ as described above. Finally, the model parameters are updated to prune filters from the layer that leads to the lowest relative reconstruction error. Algorithm 2 line 9 describes the first step, and lines 10 - 15 describe an efficient implementation of the second step, where errors for all the layers are computed in one forward pass per example. The iterations continue till an overall pruning criterion, e.g. parameter pruning ratio or percentage FLOP reduction is reached. The parameter $\alpha$ is chosen to speed up the overall execution and can be chosen as 1 if the running time is not a concern. The overall time complexity of algorithm 2, when using algorithm 1 as the filter pruning algorithm is: $\mathcal{O}(TC(N + n^4))$, where $T$ is the number of iterations needed to achieve the desired pruning (depends on $\alpha$ and the pruning criteria), $C$ is the number of layers, $N$ is the training dataset size, and $n$ is the number of filters in each layer. While `HBGS` (Algorithm 2) can select a variable number of filters from each layer, the sequential search over the layers for the best filter renders this algorithm expensive. In the next section, we develop a faster filter pruning algorithm.

## 3.3 Backward Elimination Algorithm for Filter Pruning

The time complexity of the `HBGS` algorithm depends on the training set size $N$ and the average number of filters per layer $n$. In many cases, when the time complexity of the filter pruning step ($\mathcal{O}(TCn^4)$ is substantially larger than the error computation step $\mathcal{O}(TCN)$, the complexity of the filter pruning algorithm becomes substantially larger than that of the fine-tuning algorithm on the training dataset. The main problem is the OMP-based filter pruning algorithm (FP-OMP) adds one filter in each step, which is an efficient strategy if the number of filters to be selected is small, compared to the total number of filters. However, in the context of `HBGS` algorithm, FP-OMP is sequentially called many times (Algorithm 2 line-9) with decreasing number of filters to be selected each time. In this context, a backward elimination (Couvreur & Bresler, 2000; Ament & Gomes, 2021) based approach which iteratively removes the feature which causes a minimal increase in approximation error, is intuitively more appropriate. While the original backward elimination algorithm described in (Couvreur & Bresler, 2000) is $\mathcal{O}(n^4)$, a faster implementation based on block-matrix inversion was described in (Reeves, 1999), with time complexity of $\mathcal{O}(n^3)$. Here, we derive a similar algorithm for our problem.

For simplicity, we follow the notation in (Reeves, 1999). For a given layer with $n$ output channels, $m$ input channels, and $K \times K$ filter size, we re-define the input filter matrix as $A \in \mathbb{R}^{K^2 m \times n}$, where each column is a flattened vector of filter weights, $A_{:,j} = f_{:,j}, j = 1, ..., n$. We also denote the output of the sparse approximation as $B \in \mathbb{R}^{K^2 m \times n}$, which is the same as $A$ in this case. We are interested in the approximation $B \approx A\lambda, \lambda \in \mathbb{R}^{n \times n}$, where $\lambda_{:j}$ is the weight vector for the $j^{th}$ output filter. We note that the least square

---

**Algorithm 3** Filter Pruning-Backward Elimination (`FP-Backward`)

---
1: **Input:** $n$: Number of filters, $\beta$: Pruning fraction,
2: $\qquad f_{:,j} \in \mathbb{R}^{K^2 m} \quad j = 1, ....n$: Filters
3: **Initialize:** $S = \{1, ..., n\}$ $\qquad\qquad\qquad\qquad\qquad\qquad\qquad$ ▷ Set of currently retained filters
4: $B_{:,j} = [f_{:,j}]_{j=1,...,n}$ $\qquad\qquad\qquad\qquad\qquad\qquad\qquad$ ▷ Matrix of predicted filter weights
5: $A = B$ $\qquad\qquad\qquad\qquad\qquad\qquad\qquad\qquad\qquad$ ▷ Matrix of retained filter weights
6: **while** $|S| > (1 - \beta) * n$ **do**
7: $\qquad G = [A^T A]^{-1} \in \mathbb{R}^{|S| \times |S|}$
8: $\qquad$ **for** $k = 1, ..., |S|$ **do**
9: $\qquad\qquad g_k = G_{\{-k\},k}$
10: $\qquad\qquad \gamma_k = G[k, k]$
11: $\qquad\qquad d_k = A_{-k} g_k + a_k \gamma_k$
12: $\qquad\qquad u_k = \dfrac{\sum_{j=1,...,n} |d_k^T B_{:,j}|^2}{\gamma_k}$
13: $\qquad$ **end for**
14: $\qquad k^* = argmin_{k=1,...,|S|} u_k$
15: $\qquad S \longleftarrow S \setminus \{S[k^*]\}$ $\qquad\qquad\qquad\qquad\qquad\qquad$ ▷ remove original index corresponding to $k^*$
16: $\qquad A = A_{:,\{-k^*\}}$ $\qquad\qquad\qquad\qquad\qquad\qquad\qquad\qquad$ ▷ remove selected column
17: **end while**
18: Calculate $\lambda$ using equation 3
19: **Output:** Set of selected filters-$S$, $\lambda$

---

solution for $\lambda_{:,j}$ is decoupled from $\lambda_{:,j'}$ where $j \neq j'$. Hence, the least squares solution becomes:

$$\lambda_{:,j}^* = argmin_{\lambda_{:,j}} \sum_{j \in \{1,2,..,n\}} ||B_{:,j} - \sum_{l=1,...,n} A_{:,l} \lambda_{l,j}||^2 \quad = (A^T A)^{-1} A^T B_{:,j}, \qquad \forall j = 1, ..., n \tag{3}$$

Hence, the total least square error is given by:

$$E(A, B) = \sum_{j=1,...,n} ||B_{:,j} - A(A^T A)^{-1} A^T B_{:,j}||^2 \quad = \sum_{j=1,...,n} (B_{:,j}^T B_{:,j} - B_{:,j}^T A(A^T A)^{-1} A^T B_{:,j}) \tag{4}$$

We are interested in calculating the increase in $E(A, B)$ if 1 column of $A$ and the corresponding row of $\lambda$ are removed from the input. Let $A = [A_{-k}\ a_k]\Pi_k^T$, where $A_{-k}$ is the sub-matrix of $A$ after removing the $k^{th}$ column $a_k$, and $\Pi_k$ is the permutation matrix which permutes the columns of $A$ so that $k^{th}$ column is the last. We also have the following definitions of $G_k$, $g_k$, $\gamma_k$ $D_k$ and $d_k$:

$$\begin{bmatrix} G_k & g_k \\ g_k^T & \gamma_k \end{bmatrix} = \Pi_k^T (A^T A)^{-1} \Pi_k \ ; \quad \begin{bmatrix} D_k^T \\ d_k^T \end{bmatrix} = \begin{bmatrix} G_k A_{-k}^T + g_k a_k^T \\ g_k^T A_{-k}^T + \gamma_k a_k^T \end{bmatrix} \tag{5}$$

We note from equation 4, that only the second term in $E(A, B)$ is dependent on $A$. Hence, we state the following result which connects the above equations to compute the increase in the least square error for the case of *Multivariate Linear Regression.*

**Result 2.** *Given the definitions of $A_{-k}, d_k$, and $\gamma_k$ above, the following relation holds:* $\sum_j B_{:,j}^T A_{-k}(A_{-k}^T A_{-k})^{-1} A_{-k}^T B_{:,j} = \sum_j B_{:,j}^T A(A^T A)^{-1} A^T B_{:,j} - \sum_j \frac{1}{\gamma_k} |d_k^T B_{:,j}|^2$ *hence,* $E(A_{-k}, B) = E(A, B) + \sum_{j=1,...,n} \frac{1}{\gamma_k} |d_k^T B_{:,j}|^2$.

This result is a generalization of the result reported in (Reeves, 1999). For conciseness, we provide the derivation of this result in the appendix. In light of the above result, `FP-Backward` (algorithm 3) provides the steps for backward elimination-based filter pruning. Note that line 7 in algorithm 3 is the most expensive step in the while loop (lines 6 - 17), which can be estimated from $G$ in the previous time-step using block matrix inversion with the worst-case complexity of $O(n^2)$. Also, for most calls to the algorithm, the parameter $\beta$ is very low (typically $\leq 0.05$), leading to far fewer iterations of the while loop (lines 6 - 17), which can be

---

**Algorithm 4** Layer Selection: `HBGTS`

---

1: **Input:** $C$: Number of layers, $F_c$ $c = 1, ..., C$: Filters, $\mathcal{D}$: Training dataset,
2: $\qquad$ $\alpha$: Number of filters pruned in one step, $\beta$: Pruning ratio over the network
3: **Initialize:** $y_{c,0}^0 = F_c^0 * y_{c-1,0}^0$ $\forall c = 1, ..., C$, $t \leftarrow 0$
4: **while** Overall pruning ratio $< \beta$ **do**
5: $\quad$ $e_j^t = 0$ $\forall j = 0, 1, ..., C$ $\qquad\qquad\qquad\qquad\qquad\qquad$ ▷ Initialize total error for each layer pruned
6: $\quad$ $G_c^t \leftarrow$ FP-OMP$(n, \frac{\alpha}{n}, F_c^t)$ $\forall c = 1, ..., C$ where $n = |F_c^t|$
7: $\quad$ **for** $i = 1, ..., |D|$ **do**
8: $\quad\quad$ **for** $c$ in $\{1, .., C\}$ **do**
9: $\quad\quad\quad$ $y_{c,0}^t(i) = F_c^t * y_{c-1,0}^t(i)$ $\qquad\qquad$ ▷ Unpruned propagation of unpruned previous layers
10: $\quad\quad\quad$ $y_{c,1}^t(i) = G_c^t * y_{c-1,0}^t(i)$ $\qquad\qquad\quad$ ▷ Pruned propagation of unpruned previous layers
11: $\quad\quad\quad$ **for** $j = 1, ..., c - 1$ **do**
12: $\quad\quad\quad\quad$ $y_{c,j+1}^t(i) = F_c^t * y_{c-1,j}^t(i)$ $\qquad\quad$ ▷ Unpruned propagation of previous $j^{th}$ pruned layer
13: $\quad\quad\quad$ **end for**
14: $\quad\quad$ **end for**
15: $\quad\quad$ **for** $j = 1, ..., C$ **do**
16: $\quad\quad\quad$ $e_j^t = e_j^t + \frac{||y_{C,0}^t(i) - y_{C,C-j+1}^t(i)||_2}{||y_{C,0}^t(i)||_2}$ $\qquad\quad$ ▷ Error in final layer output when $j^{th}$-layer is pruned
17: $\quad\quad$ **end for**
18: $\quad$ **end for**
19: $\quad$ $c_{min} = \text{argmin}_{j=1,...,C}(e_j^t)$
20: $\quad$ Revised network params: $F_{cmin}^{t+1} = G_{cmin}^t$ ; $F_c^{t+1} = F_c^t$ $\forall c \neq cmin$
21: $\quad$ $t \leftarrow t + 1$
22: $\quad$ Run 1 epoch of finetuning for network parameters.
23: **end while**
24: **Output:** Pruned filters $F_c^t$, $\forall c = 1, ..., C$

---

assumed to be constant. Moreover, this cost goes down with iterations as the size of the $G$ matrix reduces significantly with the iterations, reaching only a small fraction for $n$. Hence, assuming a constant number of loop executions (lines 6 - 17), the overall complexity of Algorithm 3 is $O(n^2)$, which is two orders of magnitude improvement over using algorithm 1.

### 3.4 Hierarchical Backward Greedy Tree Search (`HBGTS`)

A key idea behind the hierarchical backward greedy search (`HBGS`) algorithm is to select the layer which results in a minimal relative error when pruned from. However, the prediction error of a layer's output is not always indicative of the ultimate predictive performance of the overall network. On the other hand, computing the error of the final network output involves the re-computation of changes through all the downstream layers, starting with the pruned layer. A naive implementation of this can lead to significant compute overhead since it requires $O(CN)$ forward inferences through the network for each pruning step, where $C$ is the number of layers and $N$ is the number of data points in the training set.

Algorithm 4 presents *hierarchical backward greedy tree search* (`HBGTS`), an efficient implementation for determining the best layer to prune from at every stage. A key idea here is to calculate the error in final layer output $e_j^t$, when layer $j \in \{1, ..., C\}$ is pruned, for each input training example. Hence, we need to perform only one forward pass per pruning step for each example $i \in \{1, ..., N\}$. To implement this algorithm, we utilize a data structure $y_{c,j}^t$ which stores the output of $c^{th}$ layer $c = 1, ..., C$, when the $(n - j + 1)^{th}$ layer is pruned $j = 1, ..., C$, in the $t^{th}$ pruning step. Here, $y_{c,0}^t$ represents the output of the $c^{th}$ layer when no filter has been pruned. There are 3 cases while calculating $y_{c,:}^t$ from $y_{c-1,:}^t$ (lines 9, 10, and 12 in algorithm 4): (1) calculation of next unpruned output $y_{c,0}^t$ (unpruned propagation of unpruned previous layers), (2) calculation of output corresponding to the current pruned layer $y_{c,1}^t$ (pruned propagation of unpruned previous layers), and (3) unpruned propagation of the pruned outputs corresponding to all the previous layers. Here, we only need to store $y_{c,j}^t$ for the current timestamp $t$. Hence the space complexity of the modified algorithm increases only by $O(nd)$ where $d$ is the output size for each layer. The overall time complexity of the layer

selection algorithm for $T$ pruning steps becomes $O(TC^2N)$. Hence, using the backward elimination strategy, the overall time complexity for the proposed algorithm `HBGTS-B` is $O(TC^2N + TCn^2)$. Comparatively, the vanilla backward search version of the algorithm, `HBGS-B` using backward elimination for filter pruning takes $O(TC^2N + TCn^2)$ time. However, as shown in the next section, `HBGTS-B` marginally outperforms `HBGS-B` in terms of the accuracy of the pruned models for a given pruning ratio.

## 4 Experimental Results

In this section, we describe the experimental setup and the datasets used. We compare the performance of the proposed pruning methods against state-of-the-art methods. Furthermore, a comprehensive examination of the working of the proposed Backward Greedy Search methods is conducted.

### 4.1 Experimental setting

**Dataset Description:** For the image classification task, we utilize three datasets: CIFAR10, CIFAR100, and Tiny-Imagenet. CIFAR10 consists of 10 classes, with a training set of 50k images and a test set of 10k images, all with a resolution of $32 \times 32$. Each class contains 5k training images and 1k test images. Similarly, CIFAR100 comprises 100 classes, with 500 training images and 100 test images per class. Tiny-Imagenet contains 200 classes and includes a total of 0.1 M images. For our experimentation purpose, we resize the original $64 \times 64$ images of Tiny-Imagenet to $224 \times 224$.

**Training Details:** Our experiments involve ResNet18, ResNet32, ResNet56, VGG16, and ResNext models, with various percentages of parameter reduction. We prune a pre-trained model and adopt the training settings from LRF (Joo et al., 2021). In contrast to LRF, where the model is fine-tuned for a single epoch after each filter removal, we fine-tune the model after pruning the entire $\beta$ fraction of filters from each layer. After the completion of pruning for the entire model, we fine-tune the pruned model for 300 epochs. For fine-tuning, we set the initial learning rate to $1e^{-2}$ with a decay rate of $1e^{-4}$. We use a step scheduler that reduces the learning rate by a factor of 10 at epoch 150. Baselines were implemented using code provided by the authors and the recommended hyperparameters were used. We also performed hyperparameter search for the number of epochs, pruning ratios, and learning rates and reproduced the best results.

**Performance Metric:** We report the *test accuracy* for various pruning methods. The *dense* model's test accuracy corresponds to the *pre-trained* model's accuracy. Additionally, we report an accuracy drop (**Acc ↓**) from the dense model. We also report the drop in parameters (**param ↓**) and FLOPs (**FLOPs ↓**) as metrics to assess the level of pruning and model efficiency. The reduction in parameters refers to the decrease in the number of parameters/weights across all retained filters. FLOPs, on the other hand, refer to the number of operations (convolutions), within the retained filters.

### 4.2 Performance Comparison: Accuracy and Efficiency

We compare our proposed pruning methods with the state-of-the-art methods in Table 1. We observe that our proposed methods (`HBGS`, `HBGTS`, `FP-Backward`, `HBGS-B`, and `HBGTS-B`) exhibit higher pruned accuracy compared to state-of-the-art methods for a comparable drop in the number of parameters. We also observe that our proposed methods consistently report a higher drop in FLOPs compared to state-of-the-art methods.

**ResNet and VGG on CIFAR-100 and Tiny-Imagenet:** Figure 2 and Table 1 provide further insights into the consistently superior performance of our proposed methods. We observe that the test accuracy of the proposed methods is consistently and significantly better than the test accuracy of the baseline methods. The fact that this difference is more pronounced in a difficult dataset (CIFAR100 and Tiny-Imagenet) further demonstrates the superiority of the proposed methods. From Figure 2, we notice that, *as the percentage of parameter reduction increases, the difference in test accuracy between our proposed methods and state-of-the-art methods also grows.* **At higher parameter reduction (≥90%)**, the proposed methods outperform existing methods by $\sim 3-8\%$ (see Figure 2). Maintaining or even improving accuracy at such high parameter reduction (>95%) is very valuable, further highlighting the effectiveness of the proposed methods.

Table 1: Performance comparison between different pruning methods on **VGG16/CIFAR100** at **98%** parameter reduction, **ResNet18/CIFAR10** at **95%** parameter reduction, **ResNet56/CIFAR10** and **ResNet32/CIFAR10** at **63%** parameter reduction, averaged over three runs. ± represents standard deviation, ↓ indicate a drop, and bold/underline denotes the first/second-best result.

| Method | VGG16/CIFAR10 @ 98% | | | | ResNet18/CIFAR10 @ 95% | | | |
|---|---|---|---|---|---|---|---|---|
| | Test Acc (%) | Acc ↓ (%) | Param ↓ (%) | FLOPs ↓ (%) | Test Acc (%) | Acc ↓ (%) | Param ↓ (%) | FLOPs ↓ (%) |
| Dense | 93.2 ± 0.01 | 0 ± 0 | - | - | 94.5 ± 0.02 | 0 ± 0 | - | - |
| Random | 86.3 ± 0.18 | 6.9 ± 0.18 | 98.0 | 89.0 | 86.3 ± 0.06 | 8.2 ± 0.06 | 93.7 | 85.0 |
| EarlyCroP-S (Rachwan et al., 2022) | 90.0 ± 0.47 | 3.2 ± 0.47 | 98.0 | 91.0 | 91.0 ± 0.52 | 3.5 ± 0.52 | 95.1 | 85.8 |
| DLRFC (He et al., 2022) | 90.1 ± 0.07 | 3.1 ± 0.07 | 97.3 | 76.9 | - | - | - | - |
| SAP (Diao et al., 2023) | - | - | - | - | 92.4 ± 0.03 | 3.1 ± 0.03 | 94.9 | 84.9 |
| PL (Chen et al., 2023) | 90.2 ± 0.02 | 3.0 ± 0.02 | 97.6 | 92.0 | - | - | - | - |
| LRF (Joo et al., 2021) | 90.3 ± 0.27 | 2.9 ± 0.27 | 97.8 | 93.0 | 91.5 ± 0.37 | 3.0 ± 0.37 | 95.1 | 85.8 |
| FP-Backward | 91.5 ± 0.08 | 1.7 ± 0.08 | 97.8 | 93.0 | 92.8 ± 0.15 | 1.7 ± 0.15 | 95.1 | 85.8 |
| HBGS | 92.7 ± 0.21 | 0.5 ± 0.21 | 98.2 | 94.5 | 93.9 ± 0.24 | 0.6 ± 0.24 | 95.3 | 86.2 |
| HBGS-B | 92.6 ± 0.19 | 0.6 ± 0.19 | 98.1 | 94.3 | 93.7 ± 0.22 | 0.8 ± 0.22 | 95.2 | 86.0 |
| HBGTS | **93.5 ± 0.25** | **-0.3 ± 0.25** | **98.6** | **94.8** | **94.7 ± 0.28** | **-0.2 ± 0.28** | **95.6** | **86.7** |
| HBGTS-B | 93.4 ± 0.22 | −0.2 ± 0.22 | 98.5 | 94.6 | 94.6 ± 0.24 | −0.1 ± 0.24 | 95.4 | 86.5 |

| Method | VGG16/CIFAR100 @ 98% | | | | ResNet18/CIFAR100 @ 95% | | | |
|---|---|---|---|---|---|---|---|---|
| | Test Acc (%) | Acc ↓ (%) | Param ↓ (%) | FLOPs ↓ (%) | Test Acc (%) | Acc ↓ (%) | Param ↓ (%) | FLOPs ↓ (%) |
| Dense | 67.1 ± 0.01 | 0 ± 0 | - | - | 68.8 ± 0.02 | 0 ± 0 | - | - |
| Random | 55.5 ± 0.16 | 11.6 ± 0.16 | 98.0 | 86.0 | 54.9 ± 0.13 | 13.9 ± 0.13 | 93.1 | 84.2 |
| EarlyCroP-S (Rachwan et al., 2022) | 62.8 ± 0.52 | 4.3 ± 0.52 | 97.9 | 88.0 | 64.1 ± 0.45 | 4.7 ± 0.45 | 94.3 | 86.5 |
| DLRFC (He et al., 2022) | 63.5 ± 0.09 | 3.56 ± 0.09 | 97.1 | 53.7 | - | - | - | - |
| PL (Chen et al., 2023) | 63.5 ± 0.03 | 3.6 ± 0.03 | 97.3 | 87.9 | - | - | - | - |
| LRF (Joo et al., 2021) | 64.0 ± 0.31 | 3.1 ± 0.31 | 97.9 | 88.0 | 65.5 ± 0.29 | 3.3 ± 0.29 | 94.6 | 87.3 |
| FP-Backward | 66.2 ± 0.11 | 0.9 ± 0.11 | 97.9 | 88.0 | 67.9 ± 0.14 | 0.9 ± 0.14 | 94.6 | 87.3 |
| HBGS | 67.3 ± 0.17 | −0.2 ± 0.17 | 98.3 | 89.6 | 69.1 ± 0.19 | −0.3 ± 0.19 | 95.2 | 88.5 |
| HBGS-B | 67.2 ± 0.15 | −0.1 ± 0.15 | 98.1 | 89.4 | 68.9 ± 0.16 | −0.1 ± 0.16 | 95.1 | 88.3 |
| HBGTS | **67.8 ± 0.23** | **-0.7 ± 0.23** | **98.5** | **89.8** | **69.7 ± 0.26** | **-0.9 ± 0.26** | **95.4** | **88.6** |
| HBGTS-B | 67.6 ± 0.21 | −0.5 ± 0.21 | 98.4 | 89.7 | 69.5 ± 0.23 | −0.7 ± 0.23 | 95.3 | 88.5 |

| Method | ResNet56/CIFAR10 @ 63% | | | | ResNet32/CIFAR10 @ 63% | | | |
|---|---|---|---|---|---|---|---|---|
| | Test Acc (%) | Acc ↓ (%) | Param ↓ (%) | FLOPs ↓ (%) | Test Acc (%) | Acc ↓ (%) | Param ↓ (%) | FLOPs ↓ (%) |
| Dense | 93.45 ± 0.02 | 0 ± 0 | - | - | 92.49 ± 0.01 | 0 ± 0 | - | - |
| SFP (He et al., 2018) | 92.91 ± 0.47 | 0.54 ± 0.47 | 63.19 | 52.60 | 91.94 ± 0.12 | 0.55 ± 0.12 | 63.02 | 41.50 |
| FPGM (He et al., 2019) | 93.14 ± 0.21 | 0.31 ± 0.21 | 63.21 | 52.60 | 91.79 ± 0.94 | 0.70 ± 0.94 | 63.14 | 53.20 |
| HRank (Lin et al., 2020) | 92.56 ± 0.05 | 0.89 ± 0.05 | 63.04 | 62.43 | - | - | - | - |
| LFPC (He et al., 2020) | 92.89 ± 0.17 | 0.56 ± 0.17 | 63.25 | 52.90 | 91.98 ± 0.06 | 0.51 ± 0.06 | 63.05 | 52.60 |
| CHIP (Sui et al., 2021) | 92.88 ± 0.18 | 0.57 ± 0.18 | 63.12 | 62.08 | - | - | - | - |
| ASyminchange (El Halabi et al., 2022) | 93.27 ± 0.11 | 0.18 ± 0.11 | 63.28 | 62.34 | - | - | - | - |
| LRF (Joo et al., 2021) | 93.49 ± 0.13 | −0.04 ± 0.13 | 63.35 | 62.56 | 92.52 ± 0.16 | −0.03 ± 0.16 | 63.34 | 62.55 |
| FP-Backward | 93.88 ± 0.06 | −0.43 ± 0.06 | 63.35 | 62.56 | 92.85 ± 0.04 | −0.36 ± 0.04 | 63.34 | 62.55 |
| HBGS | 94.15 ± 0.09 | −0.70 ± 0.09 | 63.87 | 64.91 | 93.06 ± 0.07 | −0.57 ± 0.07 | 63.65 | 64.88 |
| HBGS-B | 94.12 ± 0.08 | −0.67 ± 0.08 | 63.72 | 64.80 | 93.03 ± 0.05 | −0.54 ± 0.05 | 63.63 | 64.78 |
| HBGTS | **94.38 ± 0.13** | **-0.93 ± 0.13** | **63.93** | **64.95** | **93.28 ± 0.12** | **-0.79 ± 0.12** | **63.76** | **64.92** |
| HBGTS-B | 94.35 ± 0.11 | −0.90 ± 0.11 | 63.89 | 64.93 | 93.26 ± 0.09 | −0.77 ± 0.09 | 63.75 | 64.90 |

| Method | ResNet56/CIFAR100 @ 98% | | | | ResNet32/CIFAR100 @ 98% | | | |
|---|---|---|---|---|---|---|---|---|
| | Test Acc (%) | Acc ↓ (%) | Param ↓ (%) | FLOPs ↓ (%) | Test Acc (%) | Acc ↓ (%) | Param ↓ (%) | FLOPs ↓ (%) |
| Dense | 69.18 ± 0.01 | 0 ± 0 | - | - | 68.48 ± 0.02 | 0 ± 0 | - | - |
| Random | 52.64 ± 0.18 | 16.54 ± 0.18 | 97.90 | 87.32 | 58.61 ± 0.16 | 9.87 ± 0.16 | 97.45 | 86.90 |
| LFPC (He et al., 2020) | 62.83 ± 0.14 | 6.35 ± 0.14 | 97.25 | 88.18 | 61.78 ± 0.13 | 6.70 ± 0.13 | 97.13 | 87.67 |
| DAIS (Guan et al., 2022) | 61.23 ± 0.16 | 7.95 ± 0.16 | 97.49 | 88.36 | 60.34 ± 0.15 | 8.14 ± 0.15 | 97.25 | 88.12 |
| GCNP (Jiang et al., 2022) | 62.71 ± 0.15 | 6.47 ± 0.15 | 97.37 | 88.42 | - | - | - | - |
| DLRFC (He et al., 2022) | 62.13 ± 0.19 | 7.05 ± 0.19 | 97.51 | 88.73 | - | - | - | - |
| LRF (Joo et al., 2021) | 63.73 ± 0.25 | 5.45 ± 0.25 | 97.84 | 88.98 | 63.17 ± 0.21 | 5.31 ± 0.21 | 97.43 | 88.56 |
| FP-Backward | 67.66 ± 0.10 | 1.52 ± 0.10 | 98.12 | 89.24 | 66.92 ± 0.07 | 1.56 ± 0.07 | 97.87 | 89.13 |
| HBGS | 68.99 ± 0.15 | 0.19 ± 0.15 | 98.34 | 89.41 | 68.25 ± 0.14 | 0.23 ± 0.14 | 98.28 | 89.35 |
| HBGS-B | 68.82 ± 0.13 | 0.36 ± 0.13 | 98.27 | 89.32 | 68.13 ± 0.11 | 0.35 ± 0.11 | 98.23 | 89.31 |
| HBGTS | **69.52 ± 0.24** | **-0.34 ± 0.24** | **98.61** | **89.74** | **68.65 ± 0.19** | **-0.17 ± 0.19** | **98.46** | **89.71** |
| HBGTS-B | 69.39 ± 0.22 | −0.21 ± 0.22 | 98.57 | 89.68 | 68.59 ± 0.16 | −0.11 ± 0.16 | 98.38 | 89.63 |

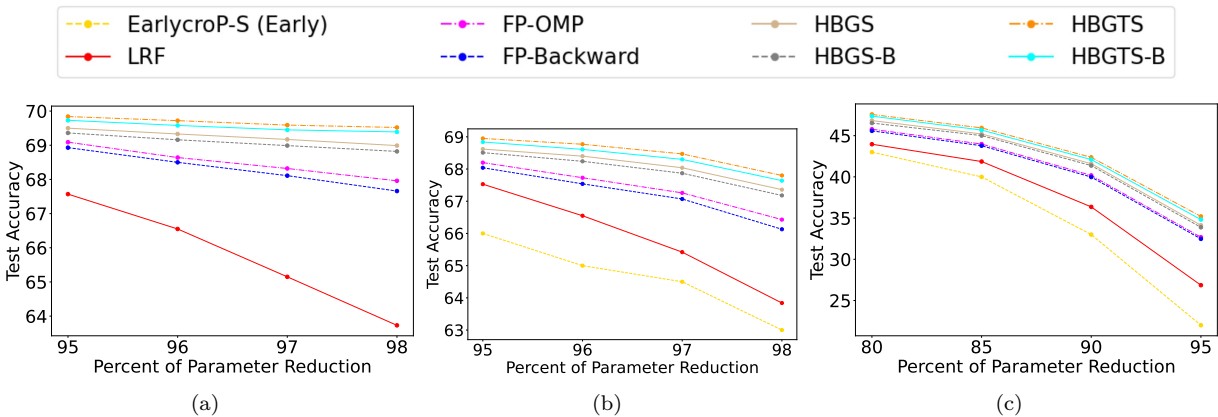

Figure 2: Test accuracy for (a) **ResNet56/CIFAR100** (b) **VGG16/CIFAR100** and (c) **ResNet18/Tiny-Imagenet** with increasing parameter reduction.

**To prune a Large Model:** Our backward greedy search methods can be used for effectively pruning large models that exceed the capacity of commodity GPUs. We use ResNext101 32x16d as our large model, consisting of 193 M parameters and requires 7.62 GB of GPU memory for loading. Additionally, we use ResNext101 32x8d as our smaller dense model, which has 88 M parameters and requires 3.91 GB for GPU memory. Table 2 shows that when ResNext101 32x16d pruned to 98% parameter reduction using `HBGTS-B`, achieves a test accuracy that matches its dense counterpart. Hence, we can efficiently deploy the pruned model on edge devices with GPU memory less than 2GB. Furthermore, the pruned model takes 5.04 times less GPU memory than the larger dense model. Notably, the pruned model even outperforms the smaller dense model, ResNext101 32x8d.

Table 2: Comparison of pruning methods for **ResNext101 32x16d** (RN16) and a similar sized dense **ResNext101 32x8d** (RN8) on **CIFAR10** at **98%** parameter reduction.

| Method | Test Acc (%) | Acc ↓ (%) | Param ↓ (%) | FLOPs ↓ (%) | VRAM (GB) |
|---|---|---|---|---|---|
| Dense RN16 | 92.1 | 0 | - | - | 7.62 |
| Dense RN8 | 91.8 | 0 | - | - | 3.91 |
| FP-Backward | 92.9 | -0.8 | 98.5 | 89.9 | 1.59 |
| HBGS-B | **93.0** | -0.9 | 98.7 | 92.1 | 1.55 |
| HBGTS-B | **93.2** | -1.1 | 98.8 | 94.3 | 1.51 |

**Time Comparison:** Figure 3(a) provides a comparison of uniform pruning methods in terms of pruning times. We can see that our proposed method `FP-Backward` is faster than the best baseline FP-OMP by a factor of 2 for a constant pruning ratio in each layer. This is also a fair comparison since the baseline methods also prune a constant fraction of filters from each layer.

Figure 3(b) compares the pruning times of various non-uniform pruning methods. Our proposed methods (`HBGS`, `HBGS-B`, `HBGTS`, and `HBGTS-B`) demonstrate superior computational efficiency compared to the baseline EarlyCroP-S, a non-uniform pruning method. Notably, `HBGS` and `HBGTS` exhibit higher pruning times relative to `HBGS-B` and `HBGTS-B`. Specifically, `HBGS-B` achieves a 54.40% and 56.16% reduction in pruning time compared to `HBGS` for ResNet32 and ResNet56, respectively. Likewise, `HBGTS-B` shows a 55.03% and 57.58% reduction in pruning time compared to `HBGTS` for ResNet32 and ResNet56, underscoring the effectiveness of the backward elimination strategy. Further, as expected `HBGTS` is computationally more expensive compared to `HBGS` with approximately double the time. The most efficient hierarchical pruning method (`HBGS-B`) takes 5 hours for ResNet32 (see Figure 3(b)) (for $\alpha = 5$, number of filters removed in each round) compared to 1 hour taken by FP-OMP. The increase in time can be further reduced by pruning a higher number of filters ($\alpha$) in each round.

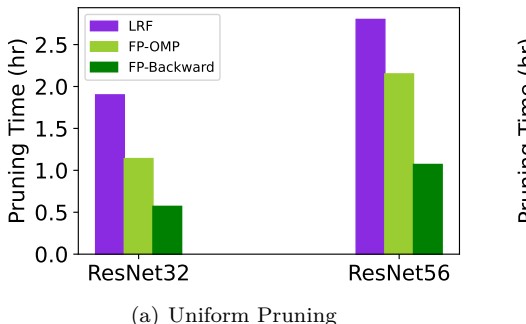

(a) Uniform Pruning

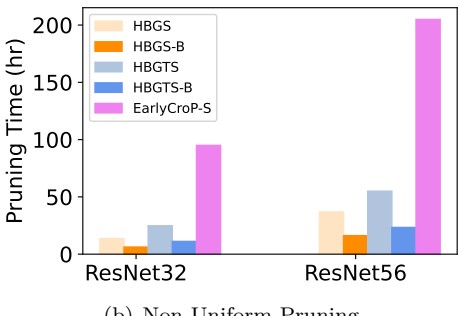

(b) Non-Uniform Pruning

Figure 3: Time comparison on **ResNet/CIFAR10** at **63%** parameter reduction.

### 4.3 Analysis of Backward Greedy Search Algorithms

We analyze the working of the proposed greedy search methods in terms of their pruning quality. Figure 4 illustrates a heat map showcasing the relative reconstruction error and the percentage of removed filters for each layer across the pruning rounds, using `HBGTS-B` method for ResNet32 on the CIFAR-100 dataset at 63% parameter reduction. The relative reconstruction error is calculated as $\frac{||y_{C,0}^t - y_{C,c}^t||_2}{||y_{C,0}^t||_2}$ where, $y_{C,0}^t$ is the output from the final classification layer when no pruning was done in any of the layers of the network and $y_{C,c}^t$ is the output from the final classification layer when pruning was done at layer c. Both the relative reconstruction error and the pruning percentage are depicted after every $7^{th}$ round, each pruning 5 filters. Examining Figure 4, we observe that the pruning percentage increases with each round, but not uniformly. For example, layers 14 - 18 have higher pruning compared to layers 1-3. Relative reconstruction error also decreases with pruning rounds but is not uniform across layers. From the heat maps, it is evident that our method selects the layer with the least relative reconstruction error for pruning. For example, layers 14 - 18 have moderate relative reconstruction errors in the initial pruning rounds, so the pruning percentage is also not so high for the same. As the pruning rounds increase, the relative reconstruction error decreases for layers 14 - 18 and hence more pruning is done from those layers as visible in the latter rounds in Figure 4. This is in contrast to uniform pruning approaches, where pruning is uniformly applied across each layer.

To understand the intuition behind the filter choices for pruning using `HBGTS-B`, we present a visualization diagram of feature maps for two layers: Layer 2 (pruned by 31.25%) and Layer 10 (pruned by 93.75%). By examining feature maps in Figure 5 (top row), we can observe that, Layer 2 exhibits a diverse range of

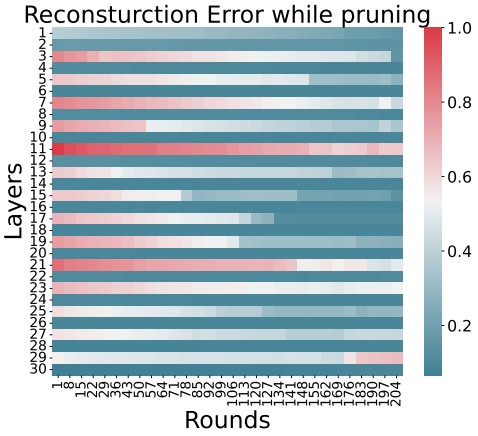

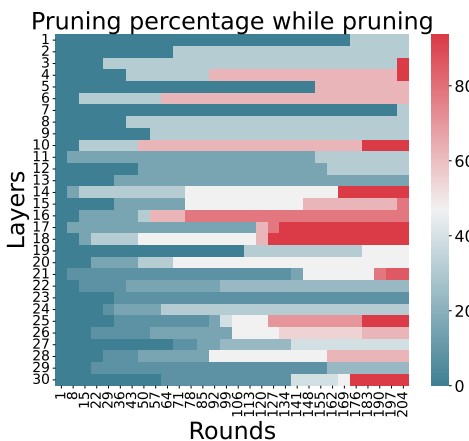

Figure 4: Heat map for relative reconstruction error and pruning percentage while pruning **ResNet32** on **CIFAR100** at **63%** parameter reduction.

31.25% pruned

93.75% pruned

Figure 5: Visualisation of output feature map of **ResNet32 2$^{nd}$ layer (top row)** and **10$^{th}$ layer (bottom row)** on **CIFAR100**

filter outputs, indicating their effectiveness in capturing various input features. Consequently, our proposed method prunes only 31.25% of the filters in Layer 2 (as shown in the last column of pruning percentages in Figure 4). Similarly, Figure 5 (bottom row) displays feature map outputs from Layer 10, which appear very similar, indicating redundancy in filter outputs. This observation aligns with the pruning percentages shown in the last column of Figure 4, where Layer 10 has 93.75% of its filters removed. Thus, we can conclude that pruning percentages yielded by `HBGTS-B` are indicative of the amount of information carried by each filter in each layer. Filters with more diverse outputs are retained, while those with redundant outputs are pruned.

## 5    Conclusion

In this paper, we propose `HBGS` and `HBGTS` as pruning methods that optimize deep neural networks by reducing the number of parameters while maintaining or even improving their accuracy. The proposed pruning methods are based on sparse approximation for non-uniform pruning, enabling the removal of redundant filters regardless of the layer they are present in. We demonstrate our method's consistent performance across various architectures, kernel sizes, and block types through extensive experiments on various datasets. We also propose `HBGS-B`, and `HBGTS-B` as efficient filter pruning methods, which offer significant advantages in terms of time efficiency compared to `HBGS`, and `HBGTS`. The proposed method can in principle be applied to any linear approximation-based pruning technique, one way of applying it to transformer-based models is pruning the filters in the feed-forward network (FFN).

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

# A  Proofs of Results

## A.1  Weight compensation for multiple channel pruning

**Result 1:** Given $Z_k$, $Z_k'$, $g_{j,k}$, $g_{l,k}'$, and $\lambda_{j,l}, j = 1, ..., n$; $l \in S$ estimated using the filter pruning process. Letting $g_{l,k}' = g_{l,k} + \sum_{l' \in S^c} \lambda_{l',l} * g_{l',k}$, $\forall l \in S$, $k = 1, ..., n$, ensures that $Z_k - Z_k' = \sum_{l' \in S^c} X * \epsilon_{l'} * g_{l',k}$, where $\epsilon_{l'}$ is the error vector for the estimation of removed filter $l' \in S^c$, and $S^c$ denotes the set of all removed filters.

Proof.  Consider the input and output of any $K \times K$ convolution layer to be $X = \{X_1, ..., X_m\}$ and $Y = \{Y_1, ..., Y_n\}$. $Y$ goes as an input to the $1 \times 1$ convolution. Let the output of the $1 \times 1$ convolution layer be $Z = \{Z_1, ..., Z_n\}$, followed by $f \in R^{m \times n}$ and $g \in R^{n \times n}$ being the filter weights of $K \times K$ and $1 \times 1$ convolution layer respectively. We can formulate the above setup as:

$$Y_j = \sum_{i=1}^m X_i * f_{i,j} := X * f_{:,j} \tag{6}$$

$$Z_k = \sum_{j=1}^n Y_j * g_{j,k} := \sum_{j=1}^n X * f_{:,j} * g_{j,k} \tag{7}$$

Now, let $f_{:,l} : l \in S$ be the selected filter weights and similarly, let $f_{:,l'} : l' \in S'$ be the pruned filter weights. Dividing Equation 7 into the two sets of filter weights, we get:

$$Z_k = \sum_{l \in S} X * f_{:,l} * g_{l,k} + \sum_{l' \in S'} X * f_{:,l'} * g_{l',k} \tag{8}$$

Following the above terminology, we can write it as:

$$f_{:,l'} = \sum_{l \in S} \lambda_{l',l} f_{:,l} + \epsilon_{l'} \quad ; \forall l' \in S' \tag{9}$$

Substituting Equation 9 in Equation 8, we rewrite $Z_k$ as $Z_k'$ in terms of retained filter weights $f_{:,l}$:

$$Z_k' = \sum_{l \in S} X * f_{:,l} * g_{l,k} + \sum_{l' \in S'} X * (\sum_{l \in S} \lambda_{l',l} f_{:,l} + \epsilon_{l'}) * g_{l',k} \tag{10}$$

The above can also be re-structured as:

$$Z_k' = \sum_{l \in S} [X * f_{:,l} * (g_{l,k} + \sum_{l' \in S'} \lambda_{l',l} * g_{l',k})] + \sum_{l' \in S'} X * \epsilon_{l'} * g_{l',k} \tag{11}$$

Once the pruning is performed, Equation 8 reduces to

$$\sum_{l \in S} X * f_{:,l} * g_{l,k} \tag{12}$$

and Equation 11 reduces to

$$\sum_{l \in S} [X * f_{:,l} * (g_{l,k} + \sum_{l' \in S'} \lambda_{l',l} * g_{l',k})] \tag{13}$$

Thus, the weight difference after pruning, for $Z_k$ and $Z_k'$, are
$\|\sum_{l' \in S'} X * f_{:,l'} * g_{l',k}\|$ and $\|\sum_{l' \in S'} X * \epsilon_{l'} * g_{l',k}\|$ respectively. Because $\epsilon_{l'} < f_{:,l'}$, the weight difference in using $Z_k'$ is lesser than that of $Z_k$. Also, the lower the difference in weights, the better the approximation.

Hence, we use Equation 13 for the weight compensation step to have a lesser weight difference and define the following step:

$$g'_{l,k} = g_{l,k} + \sum_{l' \in S'} \lambda_{l',l} * g_{l',k} \quad ; \forall k \in [1,n], \ \forall l \in S \tag{14}$$

For output channel pruning, Equation 14 is re-defined as

$$g'_{l,:} = g_{l,:} + \sum_{j \in S^c} \lambda_{j,l} * g_{j,:} \quad , \forall l \in S \tag{15}$$

while input channel pruning is re-defined as

$$g'_{:,l} = g_{:,l} + \sum_{j \in S^c} \lambda_{j,l} * g_{:,j} \quad , \forall l \in S \tag{16}$$

## A.2 Backward Elimination Algorithm for Filter Pruning

**Result 2:** Given the definitions of $A_{-k}, d_k$, and $\gamma_k$, the following relation holds: $\sum_j B_{:,j}^T A_{-k}(A_{-k}^T A_{-k})^{-1} A_{-k}^T B_{:,j} = \sum_j B_{:,j}^T A(A^T A)^{-1} A^T B_{:,j} - \sum_j \frac{1}{\gamma_k} |d_k^T B_{:,j}|^2$ hence, $E(A_{-k}, B) = E(A, B) + \sum_{j=1,\dots,n} \frac{1}{\gamma_k} |d_k^T B_{:,j}|^2$.

*Proof.* Given a matrix $A \in \mathbb{R}^{m \times n}, m \geq n$, with column rank n, and an observation matrix $B \in \mathbb{R}^{m \times j}$. The best least-squares solution to $A\lambda = B$ with at most r nonzero components is defined as the solution that minimizes the least-squares criterion.

$$Err(\lambda) = \sum_j ||B_{:,j} - A\lambda_{:,j}||_2^2 \tag{17}$$

Unconstrained least-squares solution of $A\lambda = B$ is $\lambda_{:,j} = (A^T A)^{-1} A^T B_{:,j}$. Substituting in Equation 17, we obtain

$$\begin{aligned} Err(\lambda) &= \sum_j ||B_{:,j} - A(A^T A)^{-1} A^T B_{:,j}||^2 \\ &= \sum_j (B_{:,j}^T B_{:,j} - B_{:,j}^T A(A^T A)^{-1} A^T B_{:,j}) \end{aligned} \tag{18}$$

Note that only the second term is a function of A; therefore, maximizing $B_{:,j}^T A(A^T A)^{-1} A^T B_{:,j}$ with respect to combinations of columns comprising A is equivalent to minimizing Equation 17 with respect to the combination of nonzero components in the solution.

Let $A_{-k}$ is $A$ with the $k^{th}$ column deleted. $Err_k$ can be written as

$$Err_k = \sum_j (B_{:,j}^T B_{:,j} - B_{:,j}^T A_{-k}(A_{-k}^T A_{-k})^{-1} A_{-k}^T B_{:,j}) \tag{19}$$

A simple update formula for the second term of Equation 18 can be obtained from (Reeves, 1999).

$$\begin{aligned} & \sum_j B_{:,j}^T A_{-k}(A_{-k}^T A_{-k})^{-1} A_{-k}^T B_{:,j} \\ &= \sum_j (B_{:,j}^T A(A^T A)^{-1} A^T B_{:,j} - \frac{1}{\gamma_k} |d_k^T B_{:,j}|^2) \end{aligned} \tag{20}$$

From this result, it is clear that we only need to compare $\sum_j \frac{|d_k^T B_{:,j}|^2}{\gamma_k}$ for all $k$ and eliminate the column whose corresponding value is smallest. Note that $\gamma_k$ is the $k^{th}$ diagonal element of $(A^T A)^{-1}$, and $d_k^T B_{:,j}$ is the $k^{th}$ element of the solution vector $(A^T A)^{-1} A^T B_{:,j}$. Hence we can say,

$$k^* = min_k(Err_k) \tag{21}$$

$$k^* = min_k \sum_j \frac{|d_k^T B_{:,j}|^2}{\gamma_k} \tag{22}$$

