# OpenReview forum: "A Greedy Hierarchical Approach to Whole-Network Filter-Pruning in CNNs"
_TMLR — Accepted by TMLR_

### Review · Reviewer_Dnjc · 2024-04-26

**Summary Of Contributions:**

The paper proposes filter pruning methods of CNN models leveraging the use of orthogonal matching pursuit and greedy-based approaches.  What differs from most papers in this field is that this paper proposes a way to find a different sub-optimal pruning ratio for each layer.

**Audience:**

Yes

**Broader Impact Concerns:**

There is no need to add a broader impact statement.

**Claims And Evidence:**

Yes

**Requested Changes:**

Please apply the following:

  * Page 5, Algorithm 1, line 9 -- what is $S^\prime$?
  * Page 5, Algorithm 1 and Page 8, Algorithm 4 -- $\mathcal{D}$ in Algorithm 2 is not used. While I understand the intention, do make sure to update the Algorithm to either use it explicitly or simply remove it. Same note for Algorithm 4.
  * Page 9, remove space between $0.1$ and $M$.
  * Page 9, what do mean by "we warmup the model for 20 epochs."?

**Strengths And Weaknesses:**

The paper is well-written, and I did enjoy reading it.

Strengths:
 * From a theoretical perspective, the paper does not bring a whole new theory, and most claims can be derived from the cited papers by the authors. However, using such tools does open new avenues in the field of model pruning.
 * The authors started by presenting their vanilla approaches, followed by descriptive analysis and modified approaches handling the computational time issues arising from their vanilla approaches.
 * The authors conducted experimental setting and ablation discussing and showcasing the efficacy of their approach.


Weakness:
 * See the next section for writing-related comments.
 * There are other sub-fields of channel/filter pruning that the paper did not even consider -- specifically speaking, importance-based sampling methods that have shown efficiency and efficacy; see [1], [2], [3]. Such methods usually require less than 300 epochs of fine-tuning (used in the paper) and require less time than $5$ hours of pruning. When are the proposed methods preferable to be used upon any of [1,2,3]?
 * Another line of work relies on the use of SVD which can be considered as a "distant cousin" to the OMP-based method since SVD-based approaches aim to lower the error associated with the matrix structure (lower-dimensional-based metrics for example). One paper, popping on top of my head is the [4]. Such a paper aimed also to find a suboptimal pruning parameter for each layer. How do the proposed methods compare to this paper?

-------------------------------------------
    [1] Tukan, Murad, Loay Mualem, and Alaa Maalouf. "Pruning neural networks via coresets and convex geometry: Towards no assumptions." Advances in Neural Information Processing Systems 35 (2022): 38003-38019.

    [2] Liebenwein, L., Baykal, C., Lang, H., Feldman, D., & Rus, D. (2019, September). Provable Filter Pruning for Efficient Neural Networks. In International Conference on Learning Representations.

    [3] Mussay, B., Feldman, D., Zhou, S., Braverman, V., & Osadchy, M. (2021). Data-independent structured pruning of neural networks via coresets. IEEE Transactions on Neural Networks and Learning Systems, 33(12), 7829-7841.

    [4] Liebenwein, L., Maalouf, A., Feldman, D., & Rus, D. (2021). Compressing neural networks: Towards determining the optimal layer-wise decomposition. Advances in Neural Information Processing Systems, 34, 5328-5344.

---

### Review · Reviewer_Zznz · 2024-05-20

**Summary Of Contributions:**

The paper introduces new methods for structured pruning of CNNs, called hierarchical backward greedy search (HBGS) and hierarchical backward greedy tree search (HBGTS). In each method, the authors prune entire filters using a technique similar to orthogonal matching pursuit (OMP), in which the minimize the reconstruction error of the layer's output on a given training dataset. They also add 1x1 filters to compensate for the loss. Because this is based on the reconstruction loss, the number of pruned filters can vary from one layer to another. To improve computational efficiency, teh authors also use a technique borrowed from (Reeves, 1999) using block matrix inversion to reduce the cost by two orders of magnitude from O(n^4) down to O(n^2), where n is the number of filters per layer.

Experimentally, the authors show that HBGS and HBGTS both outperform other pruning techniques in the literature using VGG16 and ResNet architectures, evaluated on CIFAR10/100 and Tiny ImageNet. In particular, the gap seems to increase as the percentage reduction in parameters increases.

**Audience:**

Yes

**Broader Impact Concerns:**

The authors should mention that pruning architectures are known to exacerbate biases in models. See for instance:

Hooker, Sara, Nyalleng Moorosi, Gregory Clark, Samy Bengio, and Emily Denton. "Characterising bias in compressed models." arXiv preprint arXiv:2010.03058 (2020).

**Claims And Evidence:**

Yes

**Requested Changes:**

- Please explain why ResNet18 results are reported for CIFAR10 only? Similarly, why are VGG16 results missing for CIFAR10?
- I think it would be useful to compare to random pruning, where you use the same pruning ratios you identify for each layer. I don't expect this to work well but it will help see the improvement that comes about by selecting the pruned filters.
- In Page 5, you claim that the algorithm is O(n^3) because |S| is small compared to n, but |S| is always proportional to n so it is O(n^4). Isn't it?

**Strengths And Weaknesses:**

*Strengths*
- The experimental results are strong. For instance, the authors achieve 94.7% on CIFAR10 using ResNet-18 after pruning more than 95% of the parameters!
- The authors include useful empirical results that support some of the rationales of their method. For example, the author show empirically (Figures 4 and 5) why layers should not have the same pruning ratio. First, layers do not have the same reconstruction error, as shown in Figure 4. Second, their filters are not of the same diversity (e.g. Layer 2 has many diverse filters but layer 10 has few). For this, HBGS prunes >90% of the filters in Layer 10 but only prunes about 1/3 of the filters in Layer 2.
- The paper is well-written.

*Weaknesses*
- My biggest concern is that it seems that the results are not reported entirely. For example, the authors report VGG16 results for CIFAR100 but not for CIFAR10. Also, they report ResNet18 results for CIFAR10 but not CIFAR100 or Tiny ImageNet. In total, the authors have experimented with 4 models and 3 datasets so we would expect a total of 12 results (for each combination) but the authors only report 4!
- The proposed algorithms are a bit complex and not easy to implement, and the authors do not indicate any plans to release their code.
- All of the experiments are done on small datasets. The largest is Tiny ImageNet which has 100K examples only. Do we observe gains when going to ImageNet-1k?

---

### Review · Reviewer_j3tq · 2024-05-30

**Summary Of Contributions:**

This paper presents an pruning approach based on several optimizations and cost reduction operations. The paper is relatively well written despite using some popularizing terms. The paper basically uses an algorithms that prunes one filter at a time as a function of output accuracy change and generalizes it to the set of $S$. In order to more efficiently search a variable groups of pruned filters in each layer, the authors introduce a backward search that simply evaluates going backward the pruning of various filters per individual layer. The filters re removed using an error reconstruction method allowing to estimate best neurons to prune. In addition in order to determine the best layer to be pruned the authors introduce a similar backward method.

**Audience:**

Yes

**Broader Impact Concerns:**

No broader impact concerns.

**Claims And Evidence:**

Yes

**Requested Changes:**

More details in the explanation and justification of the proposed method. In Tables 1 and 2 the GHBTS algorithm is the best performing by providing the most improved accuracy and pruning ratio but it is also the most time-expensive. Therefore it should be clearly explained if this model is considered as the final result and when it is worth using it. This is same in the case of the amount of pruned parameters that is in average quite low compared to previous methods.

**Strengths And Weaknesses:**

Strengths:

Well presented framework of pruning combined with backward search. The combination of the forward pruning and the backward search seems to be an efficient improvement over the original single filter pruning method.

Weaknesses:

The relative lack of novelty. While the paper is well written and the results are observable the improvement relatively minor given that the most advanced method has a magnitude larger search time when compared to the previous methods. Adding to this the relatively low improvement i wonder if the justification for such a large computation is worth such a low improvement. This should be better articulated so that specific purpose for this type of optimization is clearly observable by the reader.

---

### Decision · Action_Editor_NJBW · 2024-08-02

**Recommendation:** Accept as is

**Comment:**

While reviewers found the novelty of the material presented in this paper modest, they found the experiments solid and convincing. The topic is timely and the paper is clear. The authors also addressed the concerns raised by the reviewers in their revision, including additional experiments, adding of missing references, and providing additional discussions. All reviewers recommended acceptance.

**Audience:**

Advances in network pruning is a topic of interest to the TMLR community. While niche this work contributes to this area and provides insights into the approach proposed and methods compared to.

**Claims And Evidence:**

This work proposes a greedy pruning approach for CNNs. The method identifies the filters and the layers to be pruned via backward search strategy. The authors further propose a computationally efficient implementation of their method by building on earlier work. While somewhat niche, the proposed method is sound and the experiments show convincing evidence that the it is effective.